# Does Agroforestry Correlate with the Sustainability of Agricultural Landscapes? Evidence from China's Nationally Important Agricultural Heritage Systems

Menghan Zhang [1] and Jingyi Liu [2,*]

1   School of Landscape Architecture, Beijing Forestry University, Beijing 100083, China; zmh1993@bjfu.edu.cn
2   College of Forestry and Landscape Architecture, South China Agricultural University, Guangzhou 510642, China
*   Correspondence: liujingyi@scau.edu.cn; Tel.: +86-151-2009-1302

**Abstract:** Compared with industrial monoculture, agroforestry has been perceived as a more sustainable approach to landscape management that provides various landscape-specific benefits. However, little is known about agroforestry's influence on the comprehensive sustainability of agricultural landscapes. This study focused on the importance of agroforestry and its influence on landscape sustainability, using 118 China National Important Agricultural Heritage Systems (China-NIAHS) as cases. In each China-NIAHS, we evaluated the importance of agroforestry and the landscape's comprehensive sustainability and explored their correlation. The findings indicate that agroforestry is important in most China-NIAHS. Agroforestry's importance is strongly correlated with most sustainability indicators, including biodiversity, income diversity, resource utilization, hydrogeological preservation, and water regulation. Based on the findings, we discuss the role of agroforestry in promoting sustainability and provide suggestions for sustainable management and policymaking for agricultural landscapes on a national scale.

**Keywords:** agricultural landscape; agricultural heritage; agroforestry; landscape sustainability

## 1. Introduction

More than 10% of the earth's land surface is covered by agricultural landscapes [1]. These landscapes are shaped by agricultural practices that have been adapted to specific environments for centuries [2]. Agricultural heritage landscapes are prominent examples of the results of these traditional human–nature interactions that continue to be used today [3]. They are appreciated worldwide as being rich in interrelated natural and cultural values [4,5]. To promote the understanding and conservation of agricultural heritage landscapes, in 2002, the Food and Agriculture Organization of the United Nations (FAO) launched the Globally Important Agricultural Heritage Systems (GIAHS). In response to the GIAHS, since 2012, China's Ministry of Agriculture and Rural Affairs has engaged in a long-term project to survey and protect agricultural heritage landscapes throughout China, proposing a list of China's National Important Agricultural Heritage Systems (China-NIAHS). So far, 118 sites have been identified in the China-NIAHS (Figure 1). They are referred to as the "national treasures," or "pearls of traditional wisdom" [6].

However, the current affection for modern agricultural approaches has been threatening the sustainability of many agricultural heritage landscapes [7–9]. Given that only nine plant species account for almost two-thirds of the total crop yield of agriculture [10], industrial monoculture has been a prevailing alternative to traditional agriculture. Despite its recognized economic benefits, the spread of industrial monoculture has been criticized for its negative social and environmental impacts [11]. For example, industrial monoculture is associated with soil depletion [12], inefficiency in capturing nutrients and water [13], exotic species invasion [14], biodiversity loss [15], susceptibility to pests and diseases [16],

and deterioration of cultures and local livelihoods [13]. Furthermore, these issues appear to have been aggravated as a result of climate change and resource scarcity.

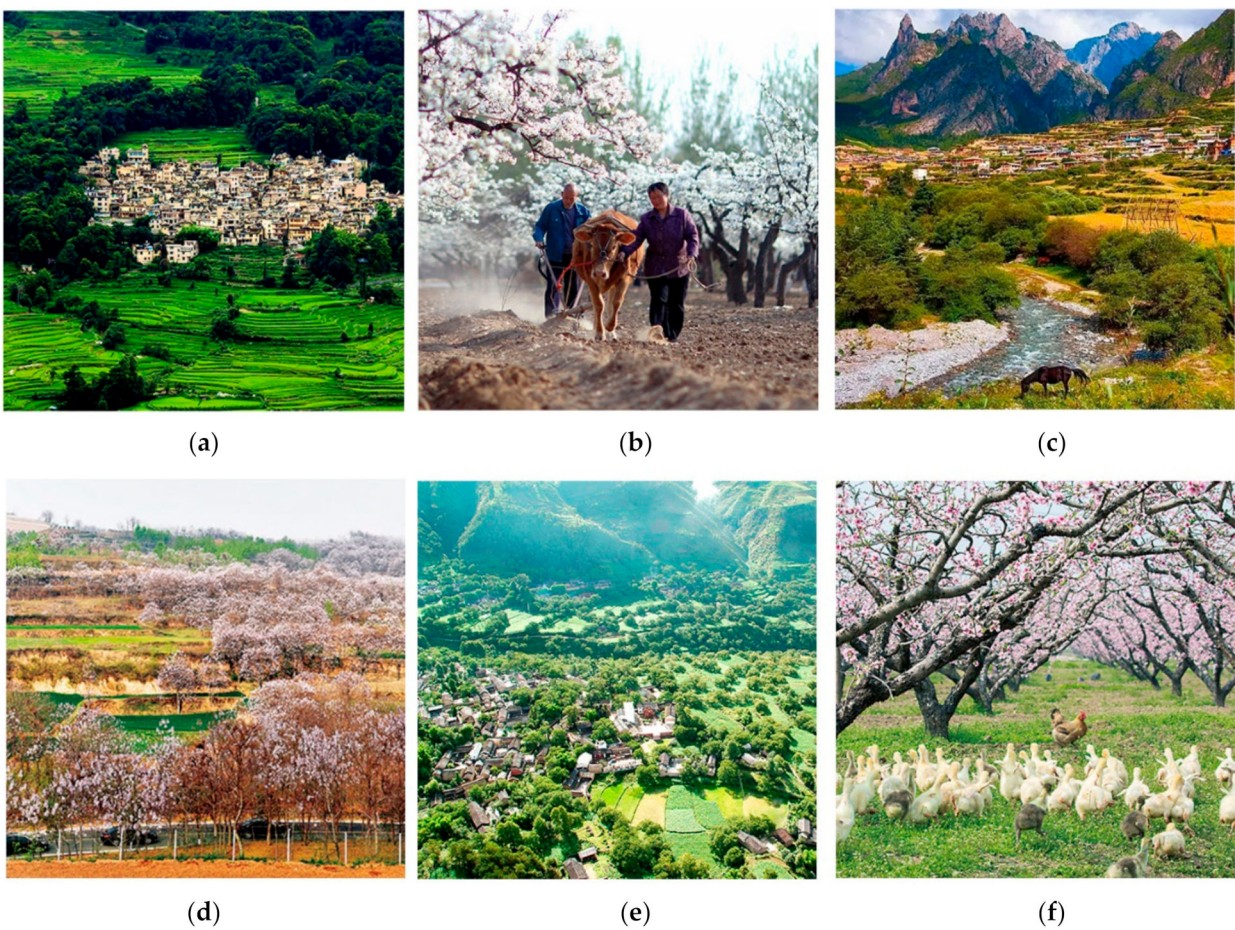

**Figure 1.** Examples of China-NIAHS that are characterized by agroforestry: (**a**) Hani Rice Terraces System; (**b**) Xiajin Ancient Mulberry Grove System; (**c**) Diebu Zhagana Agriculture-Forestry-Animal Husbandry Composite System; (**d**) Xin'an Traditional Cherry Terraces System; (**e**) Yangbi Walnut-crop Mixed System; (**f**) Yangshan Peach-Poultry Agricultural System (https://www.ciae.com.cn/list/zh/agricultural_heritage/video.html, (accessed on 22 February 2022)).

The multiple problems associated with modern agriculture prompt an investigation of the sustainability of agricultural heritage landscapes [17]. Sustainability emphasizes a long-lasting reconciliation of human development and environmental protection [18]. To understand and assess sustainability, the ecological, economic, and social benefits people obtain from landscapes have been identified [18,19]. The United Nations Sustainable Development Goals (SDG) outline a global agenda for sustainability [20]. While sustainability is relevant at multiple scales, there has been agreement that sustainability research and application to guide sustainable agricultural and rural development would be more operational at a landscape scale [21,22]. In this regard, instead of just calculating the values of assessment indicators, many studies focus on identifying the determinants shaping the sustainability of agricultural heritage landscapes [23], including those relating to labor involvement (including available laborers, intergenerational inheritance, off-farm activities, laborer's level of management experience, etc.) [24], production diversity (including biodiversity, structure of mixed crop-livestock systems, etc.) [25], landscape management (including forest conservation, irrigation systems, etc.) [26], and overall environment (including regional natural and cultural contexts, global climate change and droughts, the dynamics of public policies and local agencies, etc.) [27].

Although previous research has investigated a variety of factors that may influence landscape sustainability, the sustainability of agricultural heritage landscapes is usually assessed by the impact of different landscape management approaches [28,29]. A prominent landscape management approach characterizing agricultural heritage landscapes is agroforestry. Agroforestry, in which woody perennials are deliberately grown on the same land management units as crops and/or animals in spatial or temporal arrangements [30], is assumed to be important for the following two reasons: first, because GIAHS and China-NIAHS are named after crops, trees, and livestock, agroforestry appears to be the mainstay of the agricultural heritage landscapes. Second, agroforestry, involved in diverse landscape managements approaches, is an integral part of a sustainable working system that reconciles the values of various components (land, plant, animal, labor, irrigation, etc.) [31–33]. However, there is a lack of systematic research clearly explaining the relationship between agroforestry and sustainability compared with other determinants.

Derived from indigenous ecological practices within the context of traditional agriculture, agroforestry offers various benefits by combining the most desirable attributes of crops, trees, and livestock [34]. For example, agroforestry offers environmental benefits (including carbon sequestration, biodiversity conservation, soil enrichment, and water quality [35]), enhances local livelihoods [36], and provides social value [37]. In addition, it represents a valid means of supporting climate change mitigation while maintaining traditional systems and cultural services [38], as it entails a deep understanding of the human role in landscapes, considering indigenous knowledge and landscape management, aesthetic values, inherited cultural patterns, and people's spiritual identities [39]. Although agroforestry has been increasingly studied in recent years, most previous studies focused on individual cases, particularly on production and ecological functions. For example, He et al. (2020) studied the agrobiodiversity potential of the Shexian Dryland Stone Terraces [40]. Bai et al. (2016) investigated the water regulation of the Hani Rice Terraces [41]. Prior research has seldom, however, considered agroforestry as a sustainable approach to landscape management which embraces the complexity of socioecological systems [42]. Nor have agroforestry's effects on sustainability at a national or regional scale been examined using a larger sample size. Meanwhile, the economic and social aspects of agroforestry were frequently ignored in assessing its sustainability.

Considering agroforestry as a landscape management approach, this paper examined agroforestry on the Chinese national scale using 118 of the China-NIAHS as cases (most do not have a clear boundary, with scales varying from approximately 0.6 to 25,000 km$^2$). We explored agroforestry's importance and impact on the sustainability of agricultural heritage landscapes. Specifically, we addressed two research questions: (1) How important is agroforestry in China-NIAHS? (2) How does agroforestry influence the landscape sustainability of China-NIAHS?

To address the research questions, we collected and interpreted information about China-NIAHS from multiple sources. For each site, we measured the importance of agroforestry and evaluated landscape sustainability using multidimensional indicators. The impacts of agroforestry's importance on landscape sustainability were then examined using correlation analysis. Finally, we discussed the mechanisms of the impacts and the implications for the sustainable development of agricultural heritage landscapes at the national scale.

## 2. Materials and Methods

Four research phases were undertaken to investigate the importance of agroforestry and it impacts on the landscape sustainability of China-NIAHS (Figure 2):

1. Collection of detailed information about China-NIAHS;
2. Evaluation of the importance of agroforestry;
3. Establishment of the assessment indicator system to measure the landscape sustainability of China-NIAHS;

4.  Correlation analysis between the importance of agroforestry and the landscape sustainability of China-NIAHS.

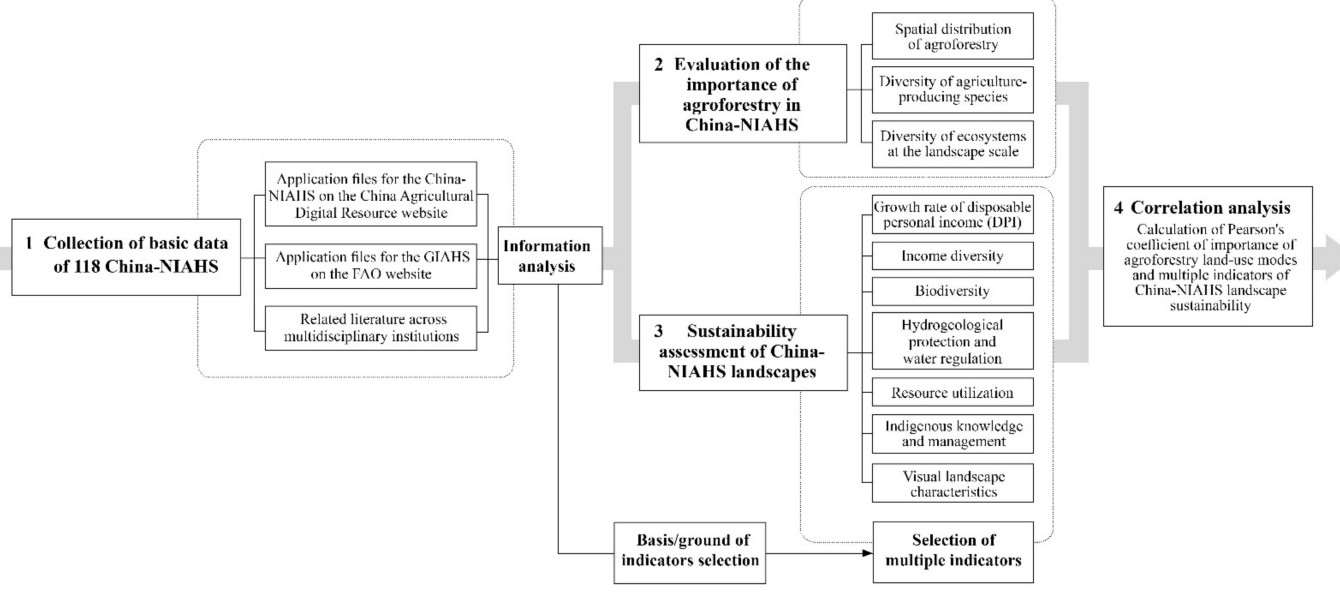

**Figure 2.** Schematic diagram of the research process.

### 2.1. Data Sources

In the first phase, a review of all the application files for the China-NIAHS (available at the China Agricultural Digital Resource website) and GIAHS (available at the FAO website) was conducted to obtain detailed information about the importance of agroforestry and its impact on landscape sustainability [43,44]. As of March 2022, there were 118 sites on the list of China-NIAHS. Besides these two authoritative platforms, we studied the literature in this field across multidisciplinary institutions (including scientific institutions, colleges, and community groups) [45]. We comprehensively analyzed the descriptive text information, research results, and photographic and video materials based on the database from those diverse sources. Considering such a wide range of sources avoids oversimplification and subjective bias, allowing for a more thorough assessment [46].

### 2.2. Evaluation of the Importance of Agroforestry in China-NIAHS

The second phase was to determine whether agroforestry is important in the 118 China-NIAHS. As importance is a value judgment that may be assessed by detailed quantitative and qualitative data, this phase primarily focused on the manual interpretation that converts the descriptive data (including numeric/text descriptions and visual information) into the indicative score. Some measures were taken to increase the objectivity of this process. First, the interpretation was guided by three explicit standards that were closely related to the importance of agroforestry based on the existing literature [47,48]: (1) spatial distribution of agroforestry (at edges, scattered, or aggregated), (2) diversity of agriculture-producing species, and (3) diversity of ecosystems at the landscape scale (farms, orchards, fish ponds, pastures, natural forests, etc.). The specific criteria for scores ranging from 1 to 5 are summarized in Table 1. Second, to reduce bias, the scoring process for each of the 118 China-NIAHS was based on the multiple information sources mentioned above as cross-references.

**Table 1.** Standard and score table of the importance of agroforestry in the China-NIAHS.

| The Criterion of the Importance of Agroforestry Land-Use Modes | | | Given Score | Importance Score |
|---|---|---|---|---|
| **Agroforestry Does not Exist in the China-NIAHS** | | | **0** | **0** |
| Agroforestry exists in the China-NIAHS | Spatial distribution of agroforestry | Only a small amount distributed at the edges | 1 | |
| | | Scattered with low density | 3 | X1 |
| | | Aggregated with high density | 5 | |
| | Diversity of agriculture-producing species | Little diversity/varieties in single agriculture (≤2) | 1 | The average of X1, X2, X3 |
| | | More diversity/varieties in single agriculture (>2) | 2 | |
| | | Involved in any two: agriculture, forestry, and husbandry, with little diversity/varieties (≤4) | 3 | X2 |
| | | Involved in any two: agriculture, forestry, and husbandry, with more diversity/varieties (>4) | 4 | |
| | | Involved in agriculture, forestry, and husbandry with more diversity/varieties | 5 | |
| | Diversity of ecosystems at the landscape scale | Involved in a small number of ecosystems (<3) | 1 | |
| | | Involved in a medium number of ecosystems (3 or 4) | 3 | X3 |
| | | Involved in a large number of ecosystems (≥5) | 5 | |

*2.3. Landscape Sustainability Assessment of China-NIAHS*

2.3.1. Indicator Selection

The third phase established an indicator system after a well-rounded study of diverse elements that reveal landscape sustainability. The FAO chose five official criteria for identifying the GIAHS: (1) food and livelihood security; (2) agrobiodiversity; (3) local and traditional knowledge systems; (4) cultural value systems and social organizations; and (5) landscape and seascape features. A survey by Santoro et al. (2020) measured the contribution of agroforestry to sustainable developments of 59 GIAHS using five indicators: (1) timber, fuelwood, food, and by-products; (2) biodiversity; (3) landscape; (4) hydrogeological protection and water regulation; and (5) structural/management characteristics [49]. Zhao et al. (2020) adopted five indicators: (1) economic contribution; (2) social equity; (3) environmental protection; (4) ecological resources; and (5) disaster resilience [50].

Based on the principles and research mentioned above, we selected seven indicators: (1) growth rate of disposable personal income (DPI); (2) income diversity; (3) biodiversity; (4) hydrogeological protection and water regulation; (5) resource utilization; (6) visual landscape characteristics; and (7) indigenous knowledge and management (Figure 3). The selection of indicators followed several principles:

(1) **Comprehensiveness**. Sustainability assessment needs to consider multiple factors that reflect the needs of humans [32,42]. The Sustainable Development Goals (SDG) are part of a program designed by the United Nations to steer global development policies and funds, from 2015 to 2030, to achieve long-term social, economic, and environmental goals [51]. As illustrated in Figure 3, the seven selected indicators are well-matched with SDG indicators and cover sustainability's ecological, economic, and social dimensions, making the indicator system more comprehensive.

(2) **Validity**. The designation of China-NIAHS is the result of an examination procedure based on qualitative criteria. The seven selected indicators are well-matched with official criteria for identifying the China-NIAHS. Therefore, the system's validation is ensured by the engagement of stakeholders in this field.

(3) **Operability**. The growth rate of the DPI was taken from each county's 2020 National Economic and Social Development Bulletin. Information on other indicators was found in the text and graphical descriptions of all application files, because these files

conform to the official criteria for recognizing China-NIAHS and GIAHS. To ensure the reliability of the evaluation results, we also referred to relevant research literature.

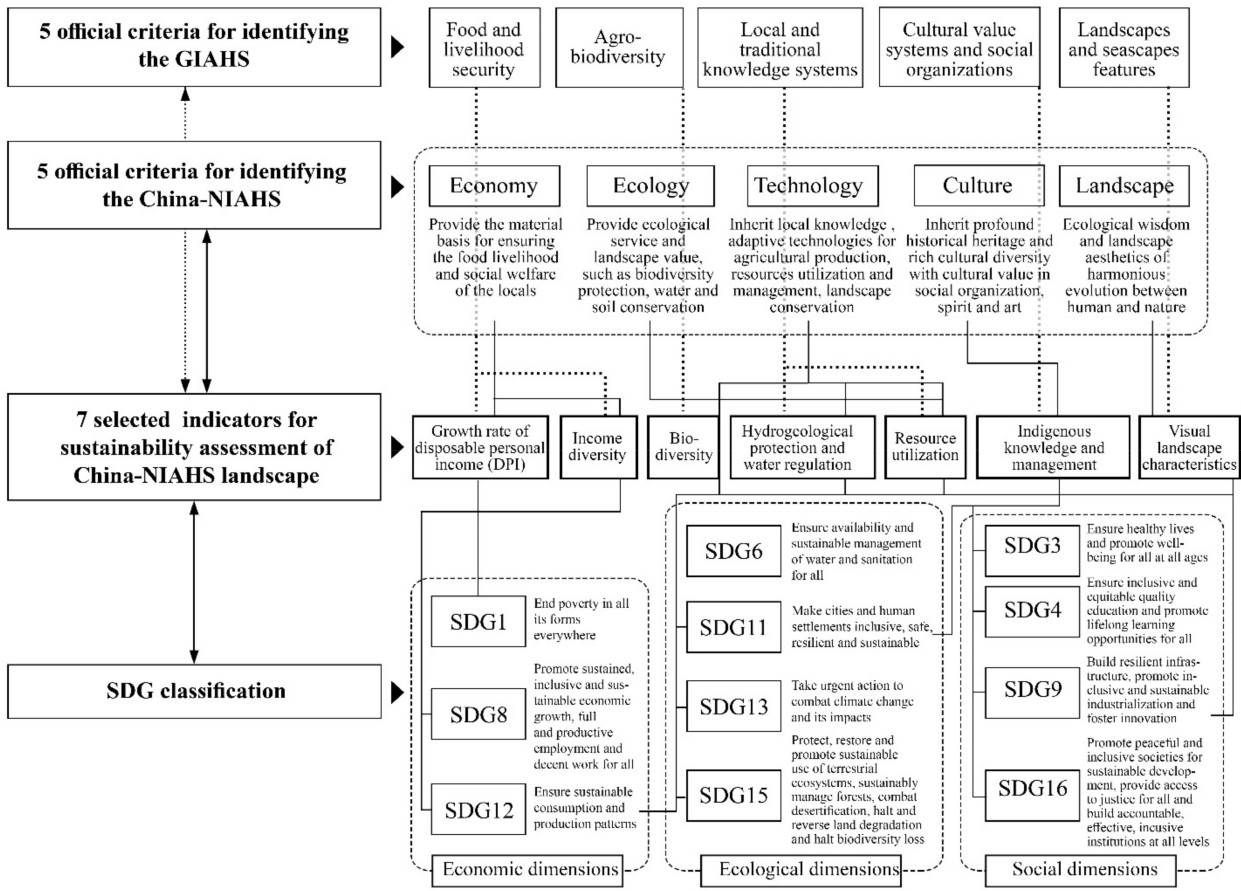

**Figure 3.** Matching diagram of landscape sustainability assessment indicators and official criteria for identifying the China-NIAHS, GIAHS, and Sustainable Development Goals.

### 2.3.2. Indicator Weight Assignment

Our assessment system for landscape sustainability uses qualitative indicators to evaluate China-NIAHS. Thus, it relies on the participation of field specialists and repeated multidisciplinary surveys. This study integrated the Delphi method (DM) with an analytic hierarchical process (AHP) to calculate the indicators' weights [52,53]. The qualitative questionnaires were distributed individually to ten specialists using the DM. The relative importance of each indicator was determined through comparison using AHP, based on qualitative questionnaires conducted by field specialists. Then, we gathered the specialists' opinions and checked for consistency (<0.10) [54]. After three rounds of anonymous consultation and feedback adjustment, a highly consistent opinion was reached (consistency test result = 0.0176). Table 2 shows the finalized sustainability assessment indicator system and weights.

**Table 2.** Indicator system of landscape sustainability assessment of the China-NIAHS.

| Goal Level | Orientation Level | Weight | Indicator Level | Content | Weight |
|---|---|---|---|---|---|
| Landscape Sustainability of China-NIAHS | Economic sustainability | 0.2099 | the growth rate of DPI | The growth rate of disposable personal income that 2020 National Economic and Social Development Bulletin collected from each county (city, district). | 0.0350 |
| | | | income diversity | Income sources that are provided by a variety of products, including but not limited to food, timber, fuelwood, fruits, herbs, fertilizers, and feed. | 0.1749 |
| | Environmental sustainability | 0.5499 | biodiversity | Biodiversity closely related to agricultural production that supports agroecosystems, including indigenous knowledge and adaptive technologies, in agricultural production and utilization of biological resources. | 0.1321 |
| | | | hydrogeological protection and water regulation | The ability to regulate water supply, improve water quality, fix soil, and protect topsoil, including indigenous knowledge and adaptive techniques in water and soil resource management and landscape conservation. | 0.3024 |
| | | | resource utilization | The degree of accessing and using environmental resources such as soil, water, light, and heat in a vertical or horizontal space, and their biological interaction. | 0.1154 |
| | Social sustainability | 0.2402 | visual landscape characteristics | The significance and protection of landscape features that reflect the ecological wisdom and landscape aesthetics of harmonious evolution between humans and nature. | 0.0400 |
| | | | indigenous knowledge and management | The scale and influence of belief and worship, cultural taboos, traditional customs, festival activities, and the management system based on indigenous knowledge inherited from previous generations. | 0.2002 |

### 2.3.3. Indicator Score Evaluation

To better understand the comprehensive landscape sustainability in each nominated area, this study assessed the degree of each indicator using primary data from the China-NIAHS, assigning a score from 0 to 3.

The indicator of the growth rate of DPI was linearly transformed to standard values, with the formula for data adjustment in the range of 0–3 being:

If $x > \sim 10\%$, $x' = 3$; If $x < 10\%$,

$$x' = \frac{x - lower(x)}{upper(x) - lower(x)} * 3$$

where, $x$ is the original data, and $x'$ is the recalibrated standard value with a threshold between 0–3. Negative values are equivalent to 0.

Except for the growth rate in DPI, we manually judged the six indicators according to application files from an authoritative platform (China Agricultural Digital Resource website) with multi-source information. This is because it was impossible to individually measure such a large number of China-NIAHS, each operating at different scales and without clear boundaries, with a comprehensive assessment. In addition to operability, several measures were taken to ensure the appropriateness of the procedure:

(1) **Collection of reliable data.** The application files we used were produced by different stakeholders (including agricultural and cooperative departments, agricultural organizations, research institutions and local governments) and examined by Chinese experts qualified in the field. Thus, the information derived from China-NIAHS application files not only reflects diverse values, but is also reliable, with less bias.

(2) **Cross-reference of multi-source information.** We verified the data by seeking multi-source information (including heritage reports, literature, websites, etc.) as a cross-reference. Such a wide range of sources can also avoid oversimplifying the complexity of agricultural heritage, allowing for a more comprehensive assessment of sustainability.

(3) **Clear and consistent scoring standard.** Rather than being a goal or condition, sustainability is viewed as a process for sustainability-enhancing decision making [55]. In response, we converted the descriptive data based on detailed China-NIAHS information into indicative data. As brief examples show in Table 3, the relative degrees were scored from 0 to 3 with a clear standard, consistent across all sites.

### 2.4. Correlation between Agroforestry and the Landscape Sustainability of China-NIAHS

In the fourth phase, Pearson's correlation coefficient ($P$) was used to analyze the degree of correlation between the scores showing the importance levels of agroforestry and the scores of multiple indicators of China-NIAHS sustainability. Pearson's correlation coefficient is one of the most common parametric tests for understanding bivariate inferential relationships [56]. Its value, which corresponds to the statistical significance of the coefficient, varies from −1 to 1, with a value closer to 1 suggesting a stronger bivariate correlation. The calculation formula is as follows:

$$P = \frac{cov(X,Y)}{\sigma_X \sigma_Y} = \frac{E((X - \mu_X)(Y - \mu_Y))}{\sigma_X \sigma_Y} = \frac{E(XY) - E(X)E(Y)}{\sqrt{E(X^2) - E^2(X)}\sqrt{E(Y^2) - E^2(Y)}}$$

where, *cov* (*X, Y*) is the covariance (an indicator reflecting the degree of correlation between two random variables), and $\sigma_X \sigma_Y$ is the standard deviation.

**Table 3.** Examples of score standard for the sustainability assessment indicator of China-NIAHS.

| Name of China-NIAHS | Interpretation of Indicator | Income Diversity I | Bio-Diversity II | Hydrogeological Protection and Water Regulation III | Resource Utilization IV | Visual Landscape Characteristics V | Indigenous Knowledge and Management VI |
|---|---|---|---|---|---|---|---|
| Huzhou Mulberry-Dyke and Fishpond System, Zhejiang | Ponds planted with mulberry trees on dykes providing leaves for silkworm rearing, with silkworm feces feeding fish. Every winter, the rich mud at the bottom floats up to the dykes as mulberry fertilizer, improving dyke soil (I, II, IV, V); making full use of water resources in low land with both high yield and adaptability to both drought and flood (I, III); indigenous knowledge survives, with sericulture folk activities and festivals flourishing, and tourism, silk, and freshwater aquaculture developed (I, VI). | 3 | 3 | 3 | 3 | 3 | 3 |
| Xinghua Duotian Agrosystem, Jiangsu | Compound production of the forest, crop, pond, and fish (I, II, IV, V); take both water storage management and flood control into account (III); unique and impressive landscape (V); indigenous knowledge survives and the tourism income is abundant (I, VI). | 2 | 2 | 3 | 2 | 3 | 3 |
| Zhangqiu Onion Cultivation System, Shandong | The main production mode of scallion and wheat rotation for more than two years (I, II, IV); traditional deep furrow Yongpei technology (VI). | 1 | 1 | 0 | 1 | 0 | 1 |
| Hami Cantaloupe Cultivation System, Xinjiang | The cultivation system of cantaloupe (I, II, IV); soil and water conservation (III); develops ecological agriculture system with tourism based on regional characteristics (VI). | 0 | 0 | 1 | 0 | 1 | 2 |

Note: The multi-indicator score of landscape sustainability: 0 = ignored, 1 = slight, 2 = obvious, and 3 = very obvious.

### 3. Results

*3.1. Importance of Agroforestry to the China-NIAHSs*

Agroforestry can be found at a total of 109 China-NIAHS (91.5%), indicating that the practice is widespread throughout China. A total of 10 sites received a score of 0 (8.5%), 15 received a score of 1~<2 (12.7%), 24 received a score of 2~<3 (20.3%), 26 received a score of 3~<4 (22.0%), and 43 received a score of 4~<5 (23.7%). In general, the importance of agroforestry is higher in southern China than in northern China (particularly in arid/semiarid northern China and in the northeast plain). Although the importance of agroforestry in China-NIAHS varies, the results indicate that agroforestry plays a fundamental role in more than half of the China-NIAHS (scores of 3~<5) (Figure 4).

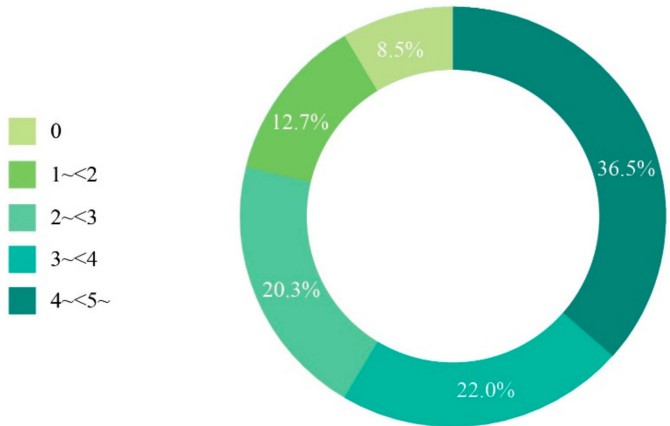

**Figure 4.** Proportion of different scores showing the importance of agroforestry in China-NIAHS.

*3.2. Correlation between the Importance of Agroforestry and the Landscape Sustainability of China-NIAHS*

Figure 5 maps the correlation between the importance of agroforestry and landscape sustainability in 118 China-NIAHS. It shows that China-NIAHS that accord greater importance to agroforestry (green solid circles with larger diameters) usually have higher landscape sustainability scores (open circles with larger diameters).

Considering the wide spatial variety in natural and cultural conditions among the 118 China-NIAHS, we analyzed the importance of agroforestry and landscape sustainability within a specific context. According to the distribution of the results in the eight agricultural zones (compiled by the National Agricultural Regionalization Committee), the landscape sustainability of China-NIAHS varies consistently with the importance of agroforestry (Figure 6). It suggests that the correlation applies to different regions of China.

Specifically, as agroforestry gains importance, most indicators and their weighted sums rise, indicating a positive correlation (Figure 7). Among these indicators, agroforestry is deemed crucial for income diversity, biodiversity, resource utilization, hydrogeological protection, and water regulation, while achieving slightly lower values for visual landscape characteristics and indigenous knowledge and management. However, it is difficult to identify the correlation between the importance of agroforestry and the growth rate of DPI.

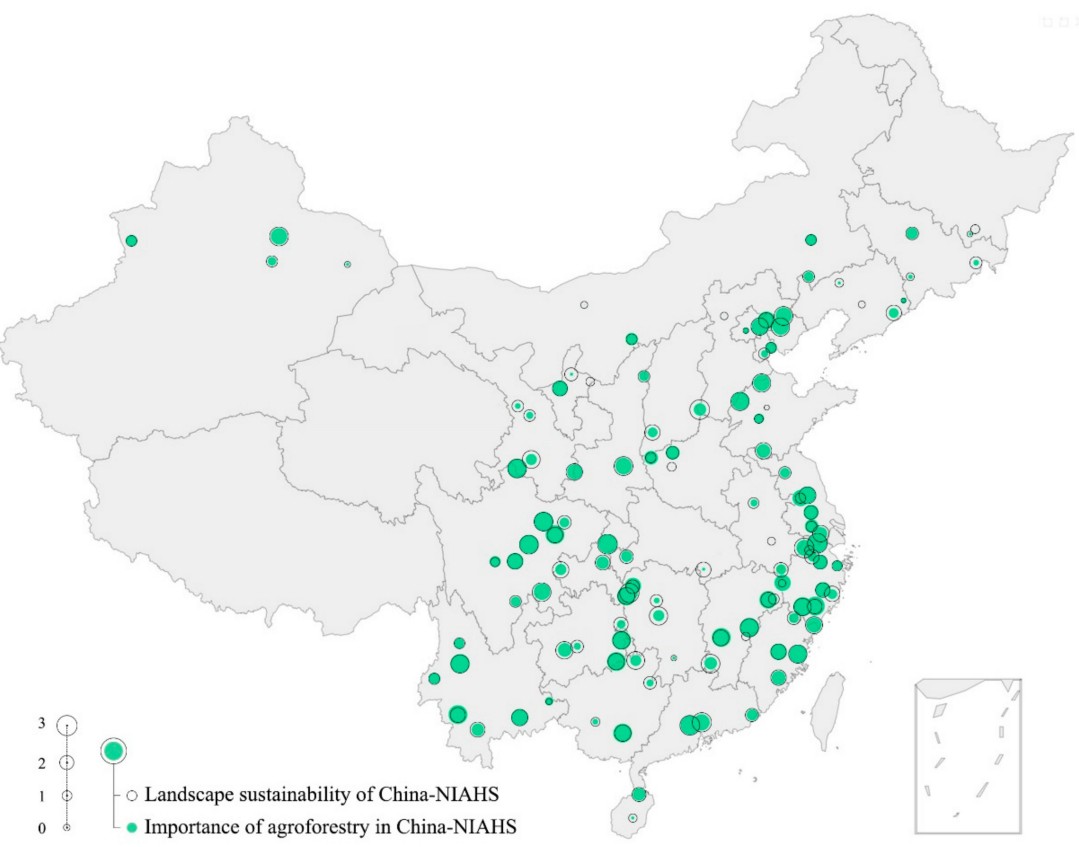

**Figure 5.** Map of the evaluation results and the relationship between the importance of agroforestry and China-NIAHS landscape sustainability.

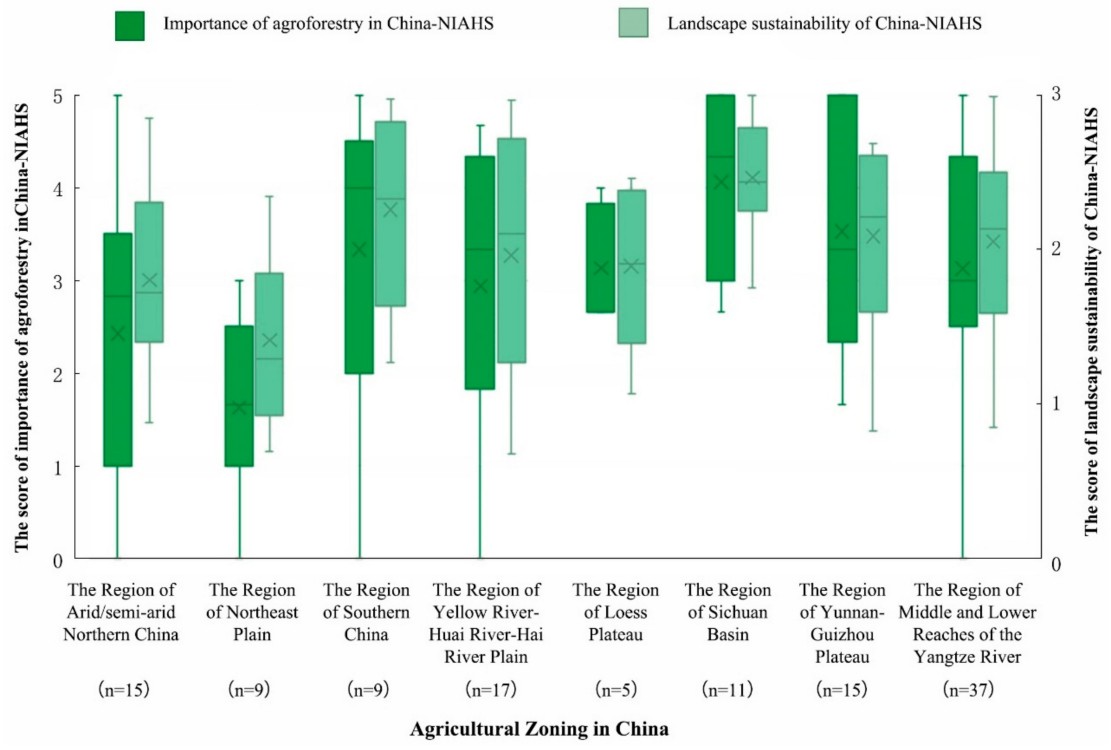

**Figure 6.** Distributions of the scores regarding the importance of agroforestry and landscape sustainability of China-NIAHS in different agricultural zones.

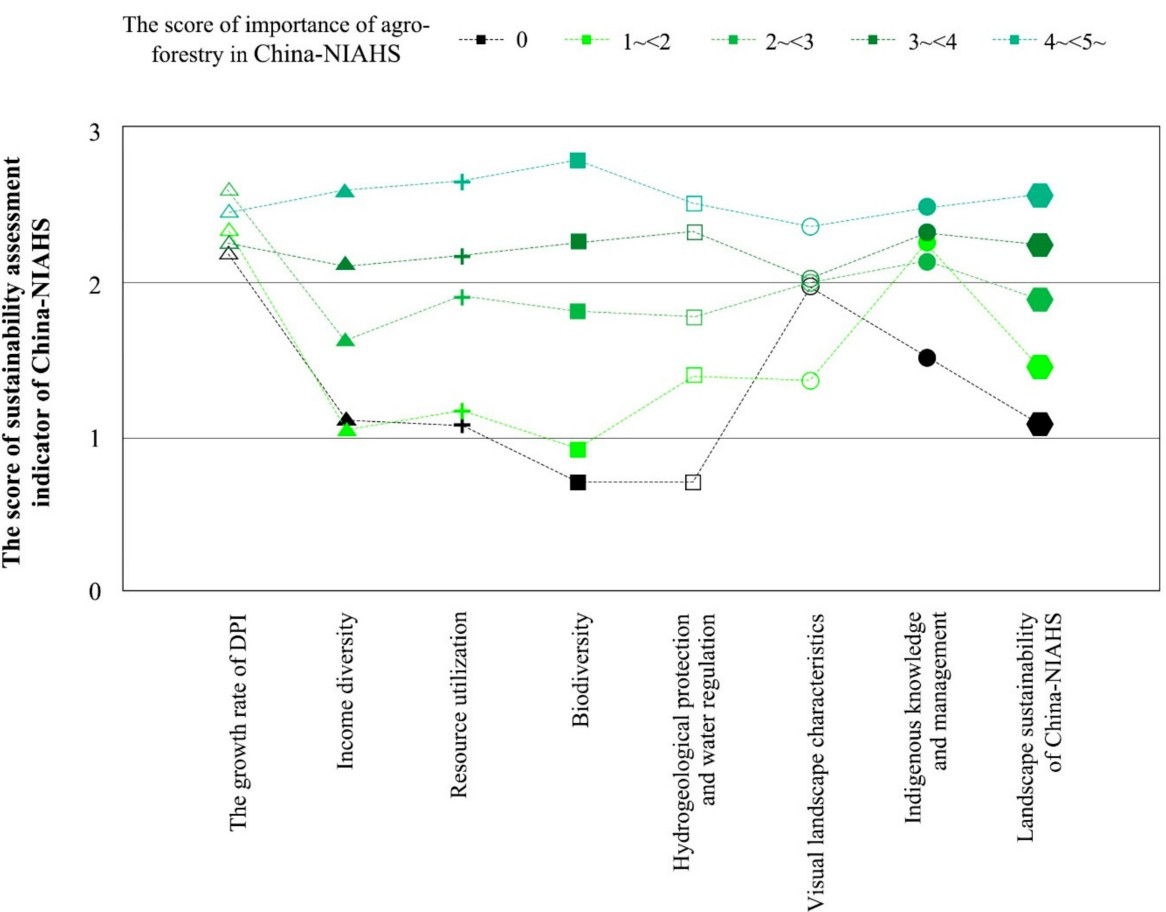

**Figure 7.** Diagram of the relationship between the importance of agroforestry and multiple indicators of China-NIAHS's landscape sustainability.

Figure 8 shows the Pearson's correlation coefficient between any two indicators. There is a statistically significant correlation between the importance of agroforestry and the weighted sum of China-NIAHS landscape sustainability ($p = 0.76$), despite a wide range of values for any two indicators ($0.02 \leq p \leq 0.94$). Four out of seven indicators suggested a strong correlation between the importance of agroforestry and the landscape sustainability of China-NIAHS due to their high coefficient values ($p \geq 0.50$). These indicators are biodiversity ($p = 0.78$), resource utilization ($p = 0.74$), hydrogeological preservation and water regulation ($p = 0.67$), and income diversity ($p = 0.52$). These findings suggest that agroforestry provides substantial ecological and economic value.

The importance of agroforestry is seen to have a medium correlation with indigenous knowledge and management (0.31) and visual landscape characteristics (0.34), as the value falls between 0.30 and 0.50. According to the findings, the social benefit that agroforestry could provide appears to be less significant than the ecological and economic benefits. However, social indicators may be more challenging to measure than ecological and economic indicators because they depend on complex, long-term processes that are difficult to identify.

Neither the importance of agroforestry nor any of the configuration indicators analyzed have a significant correlation with the growth rate of DPI ($0.02 \leq p \leq 0.10$). Compared with the strong correlation between agroforestry and income diversity ($p = 0.52$), this slight correlation ($p = 0.05$) indicates a gap between income diversity and the growth rate of DPI. This should be addressed with more ecological subsidies or mechanism adjustments, which will require joint efforts by the stakeholders in multiple fields.

Furthermore, there was a strong correlation ($p \geq 0.75$) between any combination of any two of the following—income diversity, biodiversity, and resource utilization. This correlation revealed the co-benefit of ecological and economic indicators.

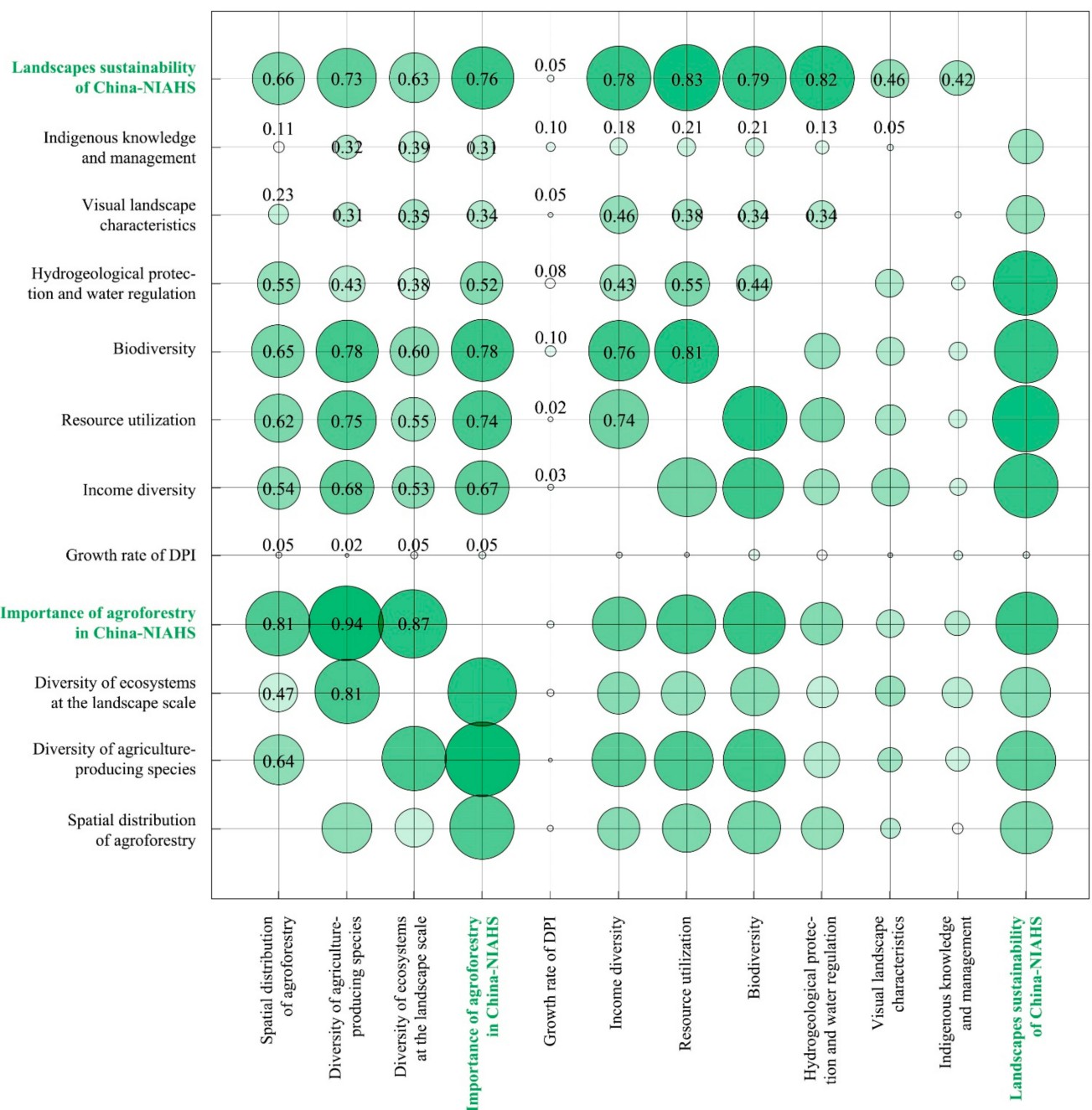

**Figure 8.** Pearson's correlation coefficient plots of the importance of agroforestry and multiple indicators of landscape sustainability. Note: Pearson's correlation coefficients are commonly classified into very high degree/strong correlation ($|p| \geq 0.75$), high degree/strong correlation ($0.50 \leq |p| < 0.75$), moderate degree/medium correlation ($0.30 \leq |p| < 0.50$), low degree/small correlation ($|p| < 0.30$), and no correlation ($p = 0$).

## 4. Discussion

### 4.1. The Specific Roles of Agroforestry in the Landscape Sustainability of China-NIAHS

The results revealed a strong correlation between the importance of agroforestry and the landscape sustainability of China-NIAHS. The analysis of the China-NIAHS dossiers

and many studies highlighted the fundamental roles that agroforestry plays in influencing landscape sustainability, which can be discussed and categorized into the following three different types.

First, agroforestry is crucial for hydrogeological protection and water regulation ($p$ = 0.52). This is consistent with previous research conducted on the GIAHS in Asia [49]. This function of agroforestry can be seen in the China-NIAHS that are characterized by steep cultivated slopes with terraced landscapes, such as the Hani Rice Terraces System (Yunnan) and the Pidu Forest Farming Culture System (Sichuan). At these sites, forests are mainly located at the upper fringes overlooking the terraces, where they can regulate the impact of precipitation and prolong the water supply. They retain water and slowly release it to the terraces below, avoiding excessive water flow. At the same time, the forest roots help to stabilize soil and prevent landslides, as in the Xiajin Ancient Mulberry Grove System (Shandong) and the Jiaxian Traditional Chinese Date Gardens (Shaanxi). In addition, some agroforestry land use has an indirect role in water resource regulation. For example, in the Wannian Traditional Rice Culture System (Jiangxi), mountain springs flow from the surrounding mountains and forests bringing nutrients (especially tree litter and soil minerals) that nurture high-quality rice crops.

Second, agroforestry optimizes resource utilization in China-NIAHS ($p$ = 0.74). According to the ecological niche theory, the distinct niches of different species in agroforestry actively complement each other in space and interaction [57]. The aboveground canopy and underground root system, in particular, may fully utilize light, water, and nutrients by generating a spatially vertical stratification [58,59]. Compared to common field farms, the Qianxi Chestnut Compound Cultivation System (Hebei) and the Pinggu Juglans Hopeiensis Production System (Beijing) enhance light energy consumption rates by 13% to 31% and water utilization rates by 10.3% to 15.2%, respectively [58,60]. In addition, the optimal resources utilization by agroforestry is also linked to the microclimate created by forests. This microclimate is conducive to the growth of cultivated crops in the undercanopy. In the Kuaijishan Ancient Chinese Torreya Community (Zhejiang) and the Pu'er Ancient Tea Garden and Tea Culture System (Yunnan), woody plants provide moderate shading and regulate temperature, light, wind speed, and humidity [58,61].

Last but not least, agroforestry plays a productive role in local communities. On the one hand, it can promote yield through the above-mentioned efficient use of environmental resources [59]. For example, the increase in yield due to the forest network of northern China is generally 25–50% [62]. On the other hand, the wide range of products it provides (such as timber, firewood, fruit, food, herbs, fodder, etc.) contributes significantly to income diversity ($p$ = 0.67). In the Diebu Zhagana Agriculture-Forestry-Animal Husbandry Composite System (Gansu), local forests (composed mainly of conifer species, including the protected *Taxus chinensis*) provide grazing for pigs, timber for house construction, firewood for heating, as well as fertilizers, such as bedding and wood ash. Additionally, medicinal herbs and around 88 species of medicinal and edible fungi are found in local forests [63]. Significantly, this income diversity contributes to residents' livelihoods. For example, in the Dazhai area of the Longji Terraced Fields in Longsheng (Guangxi), the income provided by terraced crops and forest by-products provides residents with a 97.7% higher income than non-residents [64].

### 4.2. Implications and Suggestions for China-NIAHS-Related Landscape Management and Policymaking

This study not only provides a detailed picture of China-NIAHS, but also gives insight into management strategies and policymaking. On the one hand, the findings assist in identifying China-NIAHS that lack sustainability. Most of these are located in North and Central China, where effective conservation plans should be implemented. To better protect and manage these at-risk China-NIAHS, the policy and economic pathways for rural development should vary according to the differences in economic, ecological, and social sustainability.

On the other hand, landscape sustainability has a strong correlation with the importance of agroforestry, and much more so with agriculture-producing species, as we discovered in our study. This finding supports previous studies indicating that agricultural diversity can improve landscape sustainability [65]. Considering agroforestry's multiple fundamental roles in promoting landscape sustainability, this realization should help promote agroforestry and its role as an integral part of a worldwide multifunctional working landscape. In response, agroforestry can be intensified or expanded to include more mixed species. Due to the interactions among elements, it may supply a great variety of benefits. As a result, agroforestry is more reliable than grant policies and subsidies for a single element when adopted in practice.

Agroforestry was shown to have a lower correlation with indigenous knowledge and management ($p = 0.31$) than with ecological and economic variables. This is conceivable because China-NIAHS are huge, complex systems with differing temporal dynamics. Thus, it is difficult to obtain information regarding the roles of indigenous knowledge and management, which are frequently inadequately described.

Even so, the analysis of the China-NIAHS dossiers and many studies has highlighted the widespread presence of indigenous knowledge and management, regardless of the importance assigned to agroforestry [66,67]. While some practices are slowly disappearing due to radical social and economic changes, most still survive and play a fundamental role in maximizing the sustainable use of resources [68,69], which is regulated by prohibitions on land-use tied to indigenous belief systems. For example, given that forests conserve soil and water, indigenous villages have consistently developed a belief system centered on sacred forest worship as part of their long-term cultivation of crops and habitation of settlements. Thus, logging and hunting are not permitted in forest areas managed by local communities under village regulations. These belief-based prohibitions on land-use are essential to terraced China-NIAHS, such as the Congjiang Rice-Fish-Duck System, the Diebu Zhagana Agriculture-Forestry-Animal Husbandry Composite System, and the Hani Rice Terrace System. This kind of bottom-up approach tailored to local conditions outperformed top-down rural planning regarding sustainability [70,71]. Consequently, we suggest that indigenous knowledge and management, as well as related belief systems collected from local stakeholders, could be integrated into planning and policymaking involving China-NIAHS [72].

### 4.3. Limitations and Outlook

This study built an approach to evaluating sustainability that aids in developing a trans-regional or national collaboration platform for interdisciplinary China-NIAHS research. However, limitations in terms of the data and objectivity of this study remain.

First, there is a lack of access to certain necessary data. On the one hand, we excluded some indicators for which data is not available, making the assessment system less comprehensive. On the other hand, the data for some indicators might not correspond to the actual situation. For example, the data on indigenous knowledge and management largely depends on the accessibility of information provided by the government and other organizations.

Second, interpretation of the Pearson's correlation coefficient largely depends on the context and purpose of the research. For example, there is likely to be a complex, non-linear relationship between the importance of agroforestry and the growth rate of DPI. Pearson's correlation coefficient could not entirely reflect this.

Third, even though this study was based on a thorough investigation of multi-source data, the indicator values were determined mainly by manual qualitative assessment, which is somewhat subjective. Therefore, future research requires a combination of subjective and objective methods to assign values in the evaluation system. As a prerequisite, more quantitative, site-specific spatial data that explore land-use modes and sustainability at multiple times and geographical scales should be obtained.

As this study aimed to provide general knowledge of agroforestry's effect on landscape sustainability, we did not conduct field surveys on each of the 118 China-NIAHS. Such a large number of sites challenged our attempt to initiate field-derived investigations and in-depth interviews in a traditional manner. Thus, place-specific case studies are needed in the future to gain more specific knowledge concerning the sustainability-enhancing mechanism by which agroforestry shapes the agricultural heritage landscapes.

There are only 118 sites on the China-NIAHS list. Given China's long history of agriculture, more agricultural heritage sites are waiting to be explored. Figure 4 shows that south China is home to the majority of the current China-NIAHS. However, the agricultural diversity of north China, especially northwest China, seems to have been ignored. Combining agricultural diversity and biological abundance datasets will allow more agricultural heritage sites to be designated in the future.

Additionally, since China-NIAHS represent unique regional conjunctions of landscape conservation, green economy, tourism, and cultural interchange [73,74], different contexts should be considered on a national and regional scale. Although we found a correlation between the importance of agroforestry and the landscape sustainability of China-NIAHS applied to different agricultural zoning, future research is needed to investigate spatial variability at a finer scales in response to comprehensive contexts (biological abundance, habitat condition, water system, socioeconomic situation, labor migration, distribution of traditional villages, etc.).This could help with targeted agricultural sustainability and rural planning strategies.

The current enthusiasm for a single agricultural ecosystem, without considering social sustainability, entails a certain risk [75]. This is particularly problematic if applied to agricultural heritage landscapes, given their long-lasting, dynamic interactions between humans and nature, inherited cultural patterns, and people's identities and values. Working as a technique lens for social sustainability, indigenous knowledge and management may enrich sustainability research, as they deepen the understanding of the role of humans in ecosystems. There have been calls for closer communication and cooperation between sustainability research and indigenous knowledge and management, implying a promising field for future research. As the study material shows, there is a lack of a precise and uniform approach to dealing with indigenous knowledge and management. Therefore, the first step is to standardize the description of indigenous knowledge and management of agroforestry based on a detailed scientific investigation in field-derived case studies. Thus, these nonmaterial landscape values can be measured explicitly in a qualitative, quantitative, or spatial way. This will motivate us to work toward a prosperous future, with a productive landscape and the coexistence of humans and nature.

**Author Contributions:** Conceptualization, M.Z. and J.L.; methodology, M.Z. and J.L.; software, M.Z. and J.L.; validation, M.Z. and J.L.; formal analysis, M.Z.; investigation, M.Z.; resources, M.Z.; data curation, M.Z.; writing—original draft preparation, M.Z.; writing—review and editing, M.Z. and J.L.; visualization, M.Z.; supervision, J.L.; project administration, J.L.; funding acquisition, J.L. All authors have read and agreed to the published version of the manuscript.

**Funding:** This research was funded by the National Natural Science Foundation of China, grant number 52108051.

**Institutional Review Board Statement:** Not applicable.

**Informed Consent Statement:** Not applicable.

**Data Availability Statement:** The data presented in this study are available on request from the corresponding author.

**Conflicts of Interest:** The funders had no role in the design of the study; in the collection, analyses, or interpretation of data; in the writing of the manuscript, or in the decision to publish the results.

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
