# Peer review of "Does Agroforestry Correlate with the Sustainability of Agricultural Landscapes? Evidence from China’s Nationally Important Agricultural Heritage Systems"

_sustainability, doi:10.3390/su14127239_

Round 1
Reviewer 1 Report
The sustainability of agricultural heritage systems is an important topic to be discussed and explored worldwide. The topic itself fits well into the scope of the journal. However, whether it is appropriate to just focus on agroforestry and its influence on the sustainability of agricultural landscapes deserves further consideration. Why not the irrigation systems? Why not available labors? Many factors can contribute to the sustainability of agricultural heritage sites. Please do consider the wide variety of factors before narrowing it down to one factor. In addition, there are other details to be addressed before it can become a solid paper.
- Statistical relationships do not indicate the mechanism. The study only identified certain correlations; please do not use the term “enhance”. There should be other ways to justify the enhancing mechanism.
- What is the study unit? Please clarify.
- There are a wide variety of landscape conditions and social contexts among the 118 China National Important Agricultural Heritage Systems. How did you address spatial variety embedded in regional natural and cultural contexts?
- Please clarify how to decide the scores of sustainability indicators. Who assigned the scores? Are they familiar with those 118 sites? Are they qualified? Please justify the appropriateness of the procedure.
Author Response
Dear Reviewer,
Thank you very much for your careful reading and the constructive comments concerning our manuscript entitled “Does Agroforestry Correlate with the Sustainability of Agri-cultural Landscapes? Evidence from China’s Nationally Important Agricultural Heritage Systems” (ID: sustainability-1712147).
Your comments are very valuable and very helpful for improving our work. We have studied comments carefully and tried our best to revise our manuscript accordingly.
Please see the point-by-point response to the comments in the attachment. And revised portion are marked up using the “Track Changes” function in the revised manuscript.
As moderate English changes are required, we used a professional English language editing service to polish up the language.
We very much appreciate your time and effort in reviewing the revised manuscript, and hope these revisions can address all the comments appropriately.
We deeply appreciate your consideration of the revised manuscript.
Yours sincerely,
The authors

Reviewer 2 Report
article is relevant, well written, well substantiated and overall mature. there is small comment for 26 paragraph - not every agricultural landscape is shaped by indigenous communities.
Author Response
Dear Reviewer,
Thank you very much for your careful reading and the constructive comments concerning our manuscript entitled “Does Agroforestry Correlate with the Sustainability of Agri-cultural Landscapes? Evidence from China’s Nationally Important Agricultural Heritage Systems” (ID: sustainability-1712147).
Your comments are very valuable and very helpful for improving our work. We have studied comments carefully and tried our best to revise our manuscript accordingly.
Please see the point-by-point response to the comments in the attachment. And revised portion are marked up using the “Track Changes” function in the revised manuscript.
Special thanks to your good comments. We very much appreciate your time and effort in reviewing the revised manuscript, and hope these revisions can address all the comments appropriately.
We deeply appreciate your consideration of the revised manuscript.
Yours sincerely,
The authors

Reviewer 3 Report
Outstanding research, analysis, and presentation. The scale of this study is impressive and because of this, the potential value of the findings to both China and other countries is increased. Given the scale of the study, this is an appropriate approach, however, it would be useful to have in field-derived case studies so the reader can see the agroforestry on the ground. Also images including overviews of the landscape would be useful. None the less a very impressive piece of research.
Author Response
Dear Reviewer,
Thank you very much for your careful reading and the constructive comments concerning our manuscript entitled “Does Agroforestry Correlate with the Sustainability of Agri-cultural Landscapes? Evidence from China’s Nationally Important Agricultural Heritage Systems” (ID: sustainability-1712147).
Your comments are very valuable and very helpful for improving our work. We have studied comments carefully and tried our best to revise our manuscript accordingly.
Please see the point-by-point response to the comments in the attachment. And revised portion are marked up using the “Track Changes” function in the revised manuscript.
Following the suggestion that minor spell check is required, we used a professional English language editing service to polish up the language.
Special thanks to your good comments. We very much appreciate your time and effort in reviewing the revised manuscript, and hope these revisions can address all the comments appropriately.
We deeply appreciate your consideration of the revised manuscript.
Yours sincerely,
The authors
